# Systematic review of shared decision-making interventions for people living with chronic respiratory diseases

Amy C Barradell [1,2,3] Charlotte Gerlis,[3] Linzy Houchen-Wolloff [3]
Hilary L Bekker,[4,5] Noelle Robertson,[6] Sally J Singh[1,3]

**Correspondence to**
Amy C Barradell;
ab1081@le.ac.uk

## ABSTRACT

**Objective** Shared decision-making (SDM) supports patients to make informed and value-based decisions about their care. We are developing an intervention to enable healthcare professionals to support patients' pulmonary rehabilitation (PR) decision-making. To identify intervention components we needed to evaluate others carried out in chronic respiratory diseases (CRDs). We aimed to evaluate the impact of SDM interventions on patient decision-making (primary outcome) and downstream health-related outcomes (secondary outcome).

**Design** We conducted a systematic review using the risk of bias (Cochrane ROB2, ROBINS-I) and certainty of evidence (Grading of Recommendations Assessment, Development and Evaluation) tools.

**Data sources** MEDLINE, EMBASE, PSYCHINFO, CINAHL, PEDRO, Cochrane Central Register of Controlled Trials, the International Clinical Trials Registry Platform Search Portal, ClinicalTrials.gov, PROSPERO, ISRCTN were search through to 11th April 2023.

**Eligibility criteria** Trials evaluating SDM interventions in patients living with CRD using quantitative or mixed methods were included.

**Data extraction and synthesis** Two independent reviewers extracted data, assessed risk of bias and certainty of evidence. A narrative synthesis, with reference to The Making Informed Decisions Individually and Together (MIND-IT) model, was undertaken.

**Results** Eight studies (n=1596 (of 17 466 citations identified)) fulfilled the inclusion criteria.
Five studies included components targeting the patient, healthcare professionals and consultation process (demonstrating adherence to the MIND-IT model). All studies reported their interventions improved patient decision-making and health-related outcomes. No outcome was reported consistently across studies. Four studies had high risk of bias, three had low quality of evidence. Intervention fidelity was reported in two studies.

**Conclusions** These findings suggest developing an SDM intervention including a patient decision aid, healthcare professional training, and a consultation prompt could support patient PR decisions, and health-related outcomes. Using a complex intervention development and evaluation research framework will likely lead to more robust research, and a greater understanding of service needs when integrating the intervention within practice.

**PROSPERO registration number** CRD42020169897.

## STRENGTHS AND LIMITATIONS OF THIS STUDY

⇒ The inclusion of studies from across the chronic respiratory disease population enabled us to draw conclusions about the efficacy of shared decision-making within this population when other reviews of specific respiratory conditions have been unable to.

⇒ The search criteria may have meant we missed studies which used alternative terminology for shared decision making.

⇒ We were unable to perform meta-analysis or sub-group analysis due to the heterogeneity of included studies and their outcome measures.

## INTRODUCTION

Patients living with chronic obstructive pulmonary disease (COPD) account for a significant proportion of death and disability worldwide.[1 2] Pulmonary rehabilitation (PR) is a recommended evidence-based intervention to improve the physical and psychological health of this population[3] but patients express significant barriers to accepting PR.[4 5] It is therefore proposed, healthcare professionals should seek to engage patients in informed decisions about their enrolment into a programme.[5 6]

Shared decision-making (SDM) is a core principle of personalised care, which encourages healthcare professionals to actively engage patients in healthcare decisions.[7] This style of communication requires patients and healthcare professionals to share knowledge about the health condition (ie, the lived experience and evidence-based treatment), then engage in a period of deliberation to review the pros and cons of each option for the patients' life and agree on an optimal healthcare choice.

This systematic review will inform the development and evaluation of a SDM intervention enabling healthcare professionals to support patients living with COPD make informed decisions about PR. SDM can be viewed as a complex intervention as it requires

BMJ

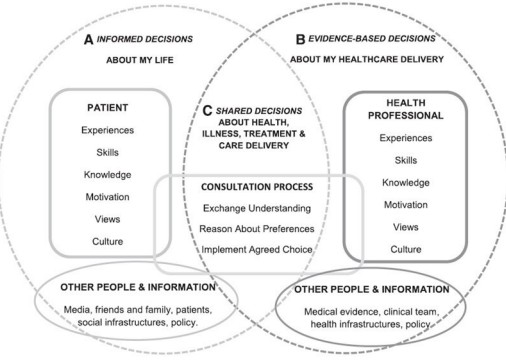

**Figure 1** The Making Informed Decisions Individually and Together model; a framework representing informed, evidence-based and shared decisions with patients and their healthcare professional.[8]

engagement from multiple stakeholders to enhance communication within a consultation. The Making Informed Decisions Individually and Together (MIND-IT) model[8] provides a conceptual framework to represent components within a multiple stakeholder decision support intervention and its evaluation (figure 1). There are a range of interventions currently supporting SDM in consultations, including patient education materials, for example, patient decision aids (PtDAs) with or without decision coaching, and SDM training for healthcare professionals.

PtDAs are resources providing accurate information to help people make informed, value-based decisions between healthcare options.[7] They are informed by decision science research to structure the content in a way that reduces the cognitive effort needed to process facts, present balanced and neutral information, and provide prompts encouraging people to trade-off their evaluations of the consequences, and choose an option that is best for their life.[9] The integration of PtDAs into multiple healthcare contexts has shown to improve the frequency of SDM and patient-centred decisions, increase patient knowledge of their condition and the options available, enhance involvement in the decision-making process and reduce decisional conflict.[10–14]

Decision coaching training for healthcare professionals aims to develop their skills in supporting patients to reason between healthcare options rather than providing information about a medical best treatment plan.[15] The training guides healthcare professionals to adopt a non-directive approach in SDM consultations by supporting patients to consider all available options, preparing them for deliberating over options and ensuring any chosen option is then implemented. Research suggests decision coaching supports patient engagement in SDM[13 16–18] and in combination with a PtDA, is a robust method to facilitate SDM.[16]

SDM training for healthcare professionals aims to help them adopt an SDM approach to delivering care.[19] There is no consensus guiding the content of this training, however, National Institute for Health and Care

Excellence offers a training package[19] which includes a range of knowledge and skills-based modules on patient-centred care, presenting risk information and uncertainty, decision and cognitive science, evidence-based medicine, multiple stakeholder collaboration and consultation skills. The evidence evaluating the effectiveness of this training on SDM use and impact in practice is weak due to the heterogeneity in study designs, methodology and evaluation strategies.[16 17]

SDM interventions are assessed using multiple outcome measures and include: within-consultation communication using healthcare professionals observation (eg, the Observer OPTION 5 tool[20]) and patient self-report questionnaires (eg, the Shared Decision-Making Questionnaire[21]), patient knowledge of healthcare options, attitudes and skills (eg, self-reported perception of risk, the Values' scale,[22] the Decision Self Efficacy Scale[23]), patient decision outcomes (eg, the choice made, the Decisional Conflict Scale,[24] the Decision Regret Scale[25]), overall SDM experience from patient and healthcare professionals perspectives (eg, the iSHARE tool[26]) and downstream patient health-related outcomes (eg, medication/treatment adherence, healthcare usage, disease knowledge and quality of life measures).

In this systematic review we sought to evaluate the efficacy of SDM interventions compared with usual care in patients living with chronic respiratory disease (CRD). To date, the efficacy of SDM interventions for adults living with CRD is largely unknown. There is a paucity of evidence to suggest patient educational materials (including PtDAs) are beneficial for reducing breathlessness and improving psychological well-being across acute respiratory diseases.[27] Two systematic reviews have been published to synthesise evidence on the impact of SDM interventions on people with asthma[28] and cystic fibrosis[29] but they found the evidence base to be very limited and too heterogeneous to draw meaningful conclusions about the relationship between the interventions and improved SDM or downstream health-related outcomes and nearly all the review population were children and adolescents. Therefore, there is a clear need to evaluate the efficacy of SDM interventions on adult patients living with CRD (eg, COPD, asthma, idiopathic pulmonary fibrosis, lung cancer).

In this review, we aim to identify the components used to support SDM within this health context and synthesise evidence of their impact on patient decision-making and downstream health-related outcomes. We use the Preferred Reporting Items for Systematic Reviews and Meta-Analyses guidelines to report this review.[30]

## METHODS
### Eligibility criteria
#### Participants
Adults (aged 18+ years) diagnosed with any CRD (and their carers).

### Intervention

Interventions which orchestrated SDM between patients and healthcare professionals, with or without disease education, decision coaching training, or SDM training were sought. For the purpose of this review, interventions needed to report a patient-facing component which was designed to facilitate the decision-making process between a patient and clinician during a consultation (eg, a patient education material).

### Comparison

Any concurrent control groups enrolled in the included studies, but not receiving an SDM intervention.

### Outcomes

Primary outcomes of interest included measures of the quality of patient decision-making (eg, participants' preparedness for decision-making, healthcare professionals and patient involvement in the decision-making process, knowledge of the available options and associated risks, decisional conflict, concordance between the participants' values and the decision made, decision confidence and decisional regret).

Secondary outcomes of interest were patients' downstream health-related outcomes (eg, behavioural activation, quality of life, knowledge of condition, healthcare usage). Specific qualitative measures of healthcare professionals' attitudes and experience of the SDM intervention, attendance, attrition, and fidelity were also included.

### Study design

Randomised and non-randomised quantitative or mixed method controlled trials were sought for inclusion in this review. This included cluster trials, interrupted time series studies with at least one data point before and after the intervention and controlled before-after studies.

Exclusions were hypothetical decision scenarios, and studies not reported in English.

### Information sources

An electronic search was conducted on the review's inception until 11 April 2023 using the following databases and registries: MEDLINE (1966–), EMBASE (1947–), PSYCHINFO (1887–), CINAHL (1937–), PEDRO (1999–), Cochrane Central Register of Controlled Trials (1993–) the International Clinical Trials Registry Platform Search Portal (2006–), ClinicalTrials.gov (2000–), PROSPERO (2011–), ISRCTN (2000–). Reference lists of selected studies and relevant systematic reviews were also searched to identify further articles for inclusion.

### Search strategy

The search algorithm (online supplemental material 1) was developed and piloted with support from the University Hospitals of Leicester NHS Trust Library Services. It included a wide array of CRDs (eg, tuberculosis, bronchiectasis, idiopathic pulmonary fibrosis). The algorithm was piloted using the MEDLINE database with limits imposed to retrieve articles in English. The algorithm was adapted to the syntax and MESH headings for each database.

### Selection process

Articles were uploaded to the web-based platform, Rayaan (https://rayyan.qcri.org/welcome) to facilitate independent review. Duplicate articles were removed and articles reporting findings across multiple outputs were highlighted to review in tandem. Two reviewers (ACB, CG) screened titles and abstracts using the inclusion and exclusion criteria. Eligible full texts were reviewed by the same two authors. Any ambiguity in the eligibility of an article was discussed and resolved with the coauthors.

### Data collection process

A review-specific data extraction form was used, informed by Cochrane data extraction forms.[31] This was piloted prior to use to ensure consistency in the extraction process. One reviewer (ACB) conducted data extraction. A random 10% selection was reviewed by a coauthor (CG) to validate the extracted data. Since there were no errors in data extraction were identified, no further validation was considered necessary. Ambiguity in the extracted data was discussed by all authors and resolved by contacting individual study teams. Specific data items for primary and subgroup analysis are outlined in online supplemental material 2.

### Study risk of bias assessment

Randomised controlled trials (RCTs) were assessed using the Cochrane Risk of Bias tool[32 33] which measures the following domains: randomisation process, deviations from intended interventions, missing outcome data, measurement of the outcomes and selection of the reported result. Non-RCTs were assessed using the Cochrane ROBINS-I tool[34] which measures the following domains: confounding, selection of study participants, classification of interventions, deviations from intended interventions, missing data, measurement of outcomes and selection of the reported result. Two authors (ACB, CG) independently rated the studies. Any discrepancy in judgements were reviewed and resolved by the coauthors.

### Certainty of evidence

The strength of evidence for all reported outcomes was assessed using the Grading of Recommendations Assessment, Development & Evaluation working group methodology (GRADE) which measures the following domains: risk of bias, inconsistency, indirectness, imprecision and publication bias. Initial judgements were based on study designs and then systematically reduced if the quality of evidence was poor or increased if a significant effect, dose response or evidence of the elimination of all plausible residual confounding variables and bias was observed. Two authors (ACB, CG) independently assessed the criteria. Any discrepancies were resolved with the coauthors.

## Synthesis methods

Statistical analysis of the results was not appropriate due to the clinical and methodological heterogeneity of the outcome measures between studies. A narrative synthesis, with reference to the MIND-IT model, was undertaken to identify the components used to support SDM in CRD and synthesise the evidence of their impact on patient decision-making and downstream health-related outcomes both between and within included studies. Guidance on the conduct of narrative synthesis in systematic reviews[35] was consulted to guide this process.

All outcomes were tabulated with continuous data presented as mean (SD) and categorical data as percentage proportions. To allow consistency in reporting and synthesis of the results, data that were not published as means and standard deviations (eg, instead reported as means and CIs) were transposed to this format using the formula reported in chapter 6 of the Cochrane Handbook for Systematic Reviews of Interventions (version 6.3, 2022[33]). These are acknowledged with an 'asterisk' in table 1.

## Patient and public involvement

No patient and public involvement occurred during the development of the systematic review's research question, the choice of outcomes measures, the design and implementation of the review, or the interpretation of the results.

## RESULTS
### Study selection

The search generated 17 488 articles. Following the removal of duplicates, 15 883 articles were retained for abstract screening. Of these, 117 underwent full text screening and 8 were retained for inclusion in this analysis (109 excluded). Most studies were excluded because of the participant population (figure 2). Other reasons included study design, type of intervention and type of publication. A full list of excluded studies is provided in online supplemental material 3.

### Study characteristics

Characteristics of the eight included studies[36–43] are provided in tables 1 and 2 (an extended version of table 1 is provided in online supplemental material 4). Six studies were RCTs,[36 38–42] two were non-RCT.[37 43] Two studies were conducted with COPD outpatients,[36 41] two with COPD inpatients,[38 40] two with lung cancer outpatients[37 43] and two with asthma outpatients.[39 42] The theoretical underpinning of interventions, reported by study authors, included social cognitive theory,[36] principles of self-management and SDM,[38] SDM,[39 42 43] self-management[40] and cognitive behavioural psychology.[41] One study[37] did not report a theoretical underpinning. The interventions of five studies included all three components of the MIND-IT multiple stakeholder model (figure 3). Four studies included additional components to SDM within their intervention groups; two included a self-management programme,[40 41] and three included an educational component.[38 39 41]

## Risk of bias within studies

Two RCTs were judged to have high risk of bias[38 41] and four RCTs were judged to have some concerns[36 39 40 42] (figure 4). One study was judged as high due to insufficient efforts in randomisation, blinding and attrition,[38] the other was due to attrition and poor fidelity.[41] The two non-RCTs were both judged to have serious risk of bias because of insufficient efforts in blinding[37 43] (figure 5).

## Results of synthesis

Results of individual studies are presented in turn alongside certainty of evidence assessments in table 1.

### Decision-making outcomes

Five outcomes captured the quality of patient decision-making (table 1); quality of communication, involvement in decision-making, decisional conflict, strength of treatment preference and knowledge of treatment options.

#### Quality of communication

One study reported that quality of communication increased in intervention and control groups, however, a significantly greater increase occurred in the intervention group alone (mean between group difference: 5.7 points; p<0.05).[36] The same study reported an increase in the occurrence of patient-centred communication in the intervention group alone, as measured by a study-specific 4-item self-reported questionnaire (mean (SD) Item 1. Intervention: 35.2 (74.6) Control: 15.9 (58.2), p<0.05; Item 2. Intervention: 60.3 (122.9) Control: 30.8 (175.5), p<0.05; Item 3. Intervention: 53.6 (115.1) Control: 45.2 (73.3), p>0.05; Item 4. Intervention: 86. 2 (84.6) Control: 75.2 (115.0), p<0.05).

#### Involvement in decision-making

Two studies reported significantly increased patient involvement in the decision-making process when compared with a control group (6 months mean (SD) between group difference: 0.4 (0.2), p<0.05; 12 months mean (SD) between group difference: 0.3 (0.1), p<0.05)[41] and a group where clinicians made treatment decisions (immediately postintervention mean (SD) between group difference: 2.5 (0.9), p<0.05).[42]

#### Decisional conflict

Two studies (one feasibility, one pilot controlled before and after study) reported a trend for reducing decisional conflict. The first reported 87% reduction in decisional conflict,[37] the other reported a mean difference of +1 indicating increased decision certainty.[43] Elsewhere, an RCT measuring disease education with and without a PtDA reported significantly reduced decisional conflict in the intervention and control groups (intervention mean(SD) difference from baseline: 8.1 (22.5), p<0.05; control mean (SD) difference from baseline: 9.1 (27.7),

**Table 1** Summary of studies

| Author, country, sample size, study design, population, healthcare decision | Outcome(s) | Summary of results | Certainty of evidence (GRADE score) |
|---|---|---|---|
| Au et al, USA,[36] clinicians=92, patients=376, RCT, outpatients with COPD, end of life care | *Validated:* Quality of communication (QOC). *Non-validated:* Occurrence of discussions about end of life preferences between patients and either clinician or surrogate | Baseline quality of communication was poor in both groups. Modest improvements in both groups, but significant mean between group difference favouring the intervention group (5.7 points; p<0.05). The occurrence (%) of patient communication about end of life care was significantly higher in the intervention group for three of four questions (mean (SD)): 1. Int: 35.2 (74.6) Con: 15.9 (58.2), p<0.05, 2. Int: 60.3 (122.9) Con: 30.8 (175.5), p<0.05, 3. Int: 53.6 (115.1) Con: 45.2 (73.3), p>0.05, 4. Int: 86.2 (84.6) Con: 75.2 (115.0), p<0.05 | ⊕⊕⊕O Moderate certainty because of risk of bias |
| Brundage et al, Canada,[37] patients=20, feasibility CBA, outpatients with lung cancer, treatment | *Validated:* Decisional conflict (DCS). *Non-validated:* Knowledge of treatment options | Decisional conflict reduced after receiving the intervention in 87% and increased in 13% of participants. Knowledge of the available treatment options improved following the intervention for both 3-year outcome survival[a] and median survival outcome[b]: 1. knowledge of survival outcomes with treatment 1 alone (mean between group difference: 60%[a], 54%[b]), 2. knowledge of direction of survival difference between treatments (mean between group difference: 27%[a], 40%[b]), 3. knowledge of the magnitude of survival difference between treatments (mean between group difference: 73%[a], 67%[b]). The number of participants with a treatment preference increased from 80% to 100% following the intervention | ⊕⊕⊕O Low certainty because of risk of bias and minor publication bias |
| Collinsworth et al, USA,[38] patients=308, RCT, inpatients with COPD, self-management | *Validated:* Patient activation (PAM), quality of life health status (CAT). *Non-validated:* All-cause hospital admissions | Both groups had a significant improvement in PAM scores following the intervention (mean (SD) difference: 0.52 (0.9) and 0.69 (1.0) points, respectively, p<0.05). Only the intervention group had significantly increased scores in the CAT following the intervention (mean (SD) difference: Int: 5.27 (10.3, p<0.05; Con: 0.38 (7.8), p>0.05). There were no significant differences in the total number of all-cause or COPD-related readmissions between groups at 1, 2, 3, 6 and 9 months (p>0.05) | ⊕⊕⊕O Moderate certainty because of risk of bias |
| Gagne et al, Canada,[39] patients=51, RCT, outpatients with asthma, treatment | *Validated:* Asthma knowledge (QCALF), decisional conflict. *Non-validated:* Appropriate use of asthma pharmacotherapy | Both groups had significantly increased knowledge scores following the intervention (mean (SD) difference: Int: 3.6 (8.9)*, p<0.001; Con: 2(7.3)*, p<0.05). There was no significant between group difference (p>0.05). Both groups had significantly reduced decisional conflict scores following the intervention (mean (SD) difference: Int: 8.1 (22.5)*, p<0.05; Con: 9.1 (27.7)*, p>0.05). There was no significant between group difference (p>0.05). Both groups had modest, but non-significant, within group improvements in their appropriate use of pharmacotherapy (mean (SD) difference: Int: 0.15 (0.7)*, p>0.05; Con: 0.16 (0.7)*, p>0.05). There was no significant between group difference (mean (SD) between group difference=0.17 (0.7), p>0.05) | ⊕⊕⊕O Moderate certainty because of risk of bias |

Continued

**Table 1** Continued

| Author, country, sample size, study design, population, healthcare decision | Outcome(s) | Summary of results | Certainty of evidence (GRADE score) |
|---|---|---|---|
| Granados-Santiago et al,[40] Spain, patients=42, RCT, inpatients with COPD, self-management | *Validated:* Quality of life and health status (EQ-5D), COPD knowledge (COPD-Q), medication adherence (TAI), physical activity (steps/day), nutritional status (MNA). *Non-validated:* – | Both groups had significant, clinically important improvements in quality of life and health status at discharge (mean (SD) between group difference: 6.15 (27.5)*, p<0.05). Only the intervention group maintained this benefit at 3 months (mean (SD) between group difference: 8.28 (50.6)*, p<0.05). Knowledge scores significantly improved in the intervention group at discharge (mean (SD) between group difference: 3.89 (0.7)*, p<0.05) and maintained at 3 months (mean (SD) between group difference: 3.88 (0.8)*, p<0.05). Adherence significantly improved at discharge in the intervention group (mean (SD) difference: Int: 3.58 (1.5)*, p<0.05; Con: 1.36 (1.6)*, p>0.05), but no significant between group difference. At 3 months, there was a significant mean (SD) between group difference favouring the intervention group (1.8 (2.4)*, p<0.05). On discharge, there was a significant reduction in physical activity in both groups (mean (SD) difference: Int: –348.27 (2.0)*, p<0.05; Con: –410.05 (4.66)*, p<0.05). At 3 months there was a significant improvement in the intervention group alone (mean (SD) difference: Int: 1371.97 (895.2)*, p<0.05; Con: 620.18 (12.5)*, p>0.05). At 3 months there was a significant improvement in nutritional status both within and between groups for the intervention group mean (SD) difference: Int: 4.15 (0.1)*, p<0.05; Con: –3.43 (0.12)*, p>0.05 | ⊕⊕⊕◯ Moderate certainty because of risk of bias |
| Myers et al, USA,[43] patients=5, pilot CBA, outpatients with non-small cell lung cancer, treatment | *Validated:* Decisional conflict (DCS SURE Test). *Non-validated:* Knowledge of treatment options, treatment status | Decisional conflict reduced postintervention (mean difference=1). Awareness of treatment options increased from 40% to all 100% participants postintervention. At 30 days postintervention, 80% of participants had made a decision. 40% of participants had a treatment that matched their treatment preference | ⊕⊕◯◯ Low certainty because of risk of bias |
| Walters et al, Australia,[41] patients=182, cluster RCT, outpatients with COPD, self-management | *Validated:* Quality of life (SF-36), quality of life (SGRQ), patient involvement in healthcare decisions (PIH). *Non-validated:* Respiratory hospital admissions, intervention fidelity | Neither group had a significant change to either their physical[a] or mental[b] health status at 6 months (mean (SD) between group difference: 1.5 (0.2)*[a], 0.8 (1.4)*[b], p>0.05) and 12 months (mean (SD) between group difference: 0.0 (0.9)*[a], 0.3 (0.9)*[b], p>0.05). Neither group had a significant change to their quality of life at 6 months (mean (SD) between group difference: 1.9 (2.7)*, p>0.05) and 12 months (mean (SD) between group difference: 1.4 (1.5)*, p>0.05. The intervention group had a significant improvement in PIH scores at 6 months (mean (SD) between group difference: 0.4 (0.2)*, p<0.05) and 12 months (mean(SD) between group difference: 0.3 (0.1)*, p<0.05). No significant change in hospital admissions for either group at 12 months ($\chi^2$=2.61, p>0.05). Fidelity assessments confirmed components were addressed with some clarity: 1. COPD symptom management, 49%, 2. unhelpful self-talk explored and identified, 37%, 3. unhelpful self-talk challenged, and new self-talk developed, 35%, 4. action plan for achieving goals made, 54%, 5. problems and barriers to achieving goals identified and clarified, 46%, 6. positive changes in behaviour praised, 83% | ⊕◯◯◯ Very low certainty because of significant risk of bias and some inconsistency |

Continued

**Table 1** Continued

| Author, country, sample size, study design, population, healthcare decision | Outcome(s) | Summary of results | Certainty of evidence (GRADE score) |
|---|---|---|---|
| Wilson et al,[42] Hawaii, patients=612, RCT, outpatients with asthma, treatment | *Validated:* Quality of life (MiniAQLQ). *Non-validated:* Healthcare use, patients' perceived role in treatment decision, medication adherence, intervention fidelity | Intervention groups had significant improvements in quality of life at 12 months (SDM to control mean (SD) between group difference=0.39 (1.0)*, p<0.05; SDM-clinician decision-making mean (SD) between group difference=0.11 (1.1)*, p>0.05; clinician decision-making-control mean (SD) between group difference=0.28 (1.0)*, p<0.05. Intervention groups had significantly reduced healthcare usage (SDM-control mean (SD) between group difference=20.36 (−1.5)*, p<0.05; SDM-clinician decision-making mean (SD) between group difference=0.01 (2.7)*, p>0.05; clinician decision-making-control mean (SD) between group difference=20.37 (−1.5*), p<0.05. The SDM group rated their influence on the treatment decision as similar to clinicians' influence (mean (SD)=3.1 (0.6). The SDM group ratings were significantly different from the clinician decision-making group, with the latter feeling their clinician had greater influence (mean (SD)=2.5 (0.9), p<0.05). Medication adherence significantly increased in intervention groups (SDM-control mean (SD) between group difference=0.21 (0.5)*, p<0.05; SDM-clinician decision-making mean (SD) between group difference=0.08 (0.4)*, p>0.05; CDM-control mean (SD) between group difference=0.13 (0.4)*, p<0.05, but not at 2 years (SDM-control mean (SD) between group difference=0.03 (0.4)*, p>0.05; SDM-clinician decision-making mean (SD) between group difference=0.04 (0.4)*, p>0.05; clinician decision-making-control mean (SD) between group difference=−0.01 (−0.4)*, p>0.05. Protocol adherence was high (SDM=4.0, clinician decision-making=3.9) and did not differ significantly between groups (p>0.05) | ⊕⊕⊕◯ Moderate certainty because of risk of bias |

3-year outcome survival[a] relates to 60%, 27% and 73%.
median survival outcome[b] relates to 54%, 40%, and 67%.

CAT, COPD Assessment Test; CBA, controlled before-after; COPD, chronic obstructive pulmonary disease; COPD-Q, Chronic Obstructive Pulmonary Disease knowledge Questionnaire; DCS, Decisional Conflict Scale; EQ-5D, EuroQol Five-Dimension; GRADE, Grading of Recommendations Assessment, Development and Evaluation; MiniAQLQ, Mini Asthma Quality of Life Questionnaire; MNA, Mini Nutritional Assessment; PAM, Patient Activation Measure; PIH, Partners in Health scale; QCALF, Questionnaire de connaissances sur l'asthme de langue française; RCT, randomised controlled trial; SDM, shared decision-making; SF-36, Short Form-36; SGRQ, St George's Respiratory Questionnaire; TAI, Test of the Adherence to Inhalers.

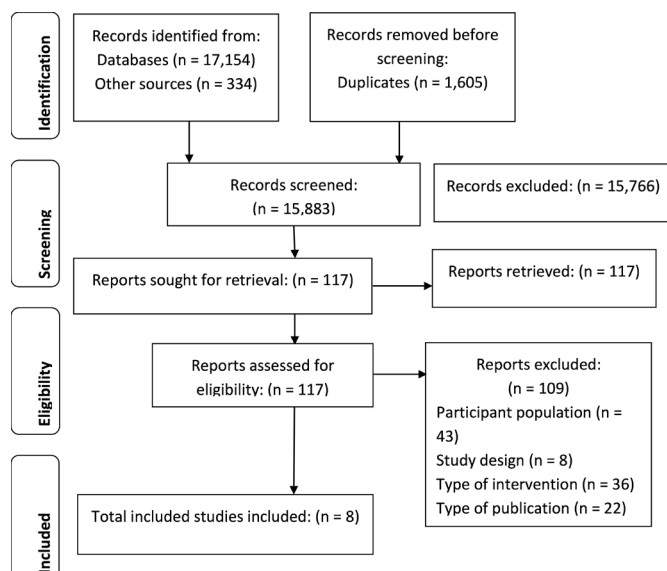

**Figure 2** Preferred Reporting Items for Systematic Reviews and Meta-Analyses flow diagram of review process.

p<0.05), but no significant mean difference between groups (between group mean difference: 2.5, p>0.05).[39]

### Strength of treatment preference

One feasibility study reported the strength of patients' treatment preference increased from 80% to 100% following exposure to a PtDA.[37]

### Knowledge of treatment options

One feasibility study reported an increasing trend for treatment specific knowledge for 3-year outcome survival[a] and median survival outcome[b] (knowledge of survival outcomes with treatment 1 alone (mean between group difference: 60%[a]; 54%[b]); knowledge of direction of survival difference between treatments (mean between group difference: 27%[a]; 40%[b]); knowledge of the magnitude of survival difference between treatments (mean between group difference: 73%[a]; 67%[b])).[37]

A pilot controlled before and after study reported awareness of treatment options increased from 40% to 100% of participants post intervention.[43]

## Health-related outcomes

Six outcomes captured downstream health-related outcomes (table 1); disease knowledge, hospitalisations, behavioural activation quality of life, intervention fidelity, attendance and attrition.

### Disease knowledge

One study reported significantly increased disease-specific knowledge scores in the intervention group (mean (SD) between group difference: 3.89 (0.7), p<0.05) which maintained at 3-month follow-up (mean (SD) between group difference: 3.88 (0.8), p<0.05).[40] However, another study reported knowledge scores increased equally among intervention and control groups (mean difference (SD) intervention: 3.6 (8.9), p<0.05; control: 2 (7.3), p<0.05).[39]

### Hospitalisations

Two studies reported no change in all-cause hospitalisations between intervention and control groups.[38 41] One study observed a significant reduction in healthcare usage in the SDM and clinician decision-making groups but not in the control group (SDM-control mean (SD) between group difference: 20.36 (1.5), p<0.05; SDM-clinician decision-making mean (SD) between group difference: 0.01 (2.7), p>0.05; clinician decision-making-control mean (SD) between group difference: 20.37 (1.5), p<0.05).[42]

### Behavioural activation

One feasibility study showed both the intervention and control group had a trend for increased levels of activation postintervention (mean (SD) between group difference: 0.52 (0.9) and 0.69 (1.0) points, respectively, p<0.05).[38]

One pilot controlled before and after study reported 80% of participants had made a treatment decision at 30 days and 40% had begun a treatment which matched their treatment preference.[43]

Two studies reported medication adherence significantly improved in the intervention groups alone.[40 42] However, the control group receiving a clinician decision-making intervention in one of these studies[42] also showed significantly adherent medication behaviour and no between group difference (SDM-clinician decision-making mean (SD) between group difference: 0.08 (0.4), p>0.05). One study measured the correct administration of medication and postintervention reported non-significant improvements in the correct use of asthma medication in the intervention and the control group with no significant difference between groups (mean (SD) between group difference: 0.17 (0.7), p>0.05).[39]

One study reported significant between group differences for improvements in nutritional behaviour at 3 months postinpatient intervention initiation (mean between group difference: 5.03; p<0.05) and physical activity at 3 months postintervention initiation (mean between group difference: 716.01; p<0.05).[40]

### Quality of life

Two studies reported no significant difference between intervention and control groups postintervention (discharge mean (SD) between group difference: 6.15 (27.5), p>0.05; 6 months mean (SD) between group difference: 1.9 (2.7), p>0.05; 12 months mean (SD) between group difference: 1.4 (1.5), p>0.05).[40 41] However, one of the studies did at 3 months postdischarge (mean (SD) between group difference: 8.28 (50.6), p<0.05).[40] While two further studies reported either a trend for improvement (mean (SD) between group difference: 4.89; p<0.05)[38] or a significant improvement in the intervention group alone (SDM to control mean (SD) between group difference: 0.39 (1.0), p<0.05).[42]

**Table 2** Summary of interventions

| Author | Description of intervention | Interventionist (including any training provided) | Intervention dose | Description of control group |
|---|---|---|---|---|
| Au et al[36] | Consultation using a completed patient-specific feedback form which describes patient preferences regarding end of life care. The feedback form was not endorsed for use | Clinicians. No specific training described | Single session | Routine outpatient COPD care |
| Brundage et al[37] | Consultation using a patient decision aid to facilitate cancer treatment decision-making | Researcher. No specific training described | Single session | Participant outcomes prior to receipt of intervention |
| Collinsworth et al[38] | COPD education and SDM self-management planning | Registered respiratory therapists. No specific training described | Single session and four telephone calls | Routine inpatient care and COPD education |
| Gagne et al[39] | Patient decision aid to facilitate asthma treatment decision-making (and asthma education) | Certified asthma educators from the Quebec Asthma and COPD Network | Single session | Routine outpatient asthma care, education and action plan generation |
| Granados-Santiago et al[40] | An SDM patient involvement programme focussing on COPD self-management goals to facilitate self-management decision-making | Healthcare professionals. No specific training described | Delivered alongside routine inpatient care | Routine inpatient care for an exacerbation of COPD |
| Myers et al[43] | Personalised patient decision aid used to facilitate 15–20 min decision coaching consultation | Oncology nurses. No training provided | Single session | Participant outcomes prior to receipt of intervention |
| Walters et al[41] | Psychoeducation about common psychological reactions to COPD diagnosis and treatment, self-management skills training, cognitive coping skills training to identify and challenge negative COPD-related cognitions, communication skills to facilitate discussion between the health mentor and the patient and promoting self-efficacy to manage chronic illness | Community health nurses, termed 'Health Mentors'. Training included 12 hours of training over 2 days that covered COPD management (1 hour), chronic disease self-management and health behaviour change components including practice role plays (7.25 hours), online training and study methods (3.75 hours) | Single session and 16 telephone calls | Routine outpatient COPD care and monthly telephone calls for 12 months (excluded intervention components) |
| Wilson et al[42] | Shared decision-making intervention: consultation to facilitate asthma treatment decision-making followed by four contacts to assess patient progress and medication changes as needed. | Healthcare professionals involved in routine asthma care (nurses, respiratory therapists, pharmacists, nurse practitioners and physician assistants). Training provided but not described. | Two sessions and three telephone calls. | Routine asthma outpatient care (including, in some sites, the opportunity to of referral to an asthma care a management programme) |
| | Clinician decision-making intervention: consultation for clinicians to obtains a patient's level of asthma control, prescribe an appropriate treatment regimen and communicate that to the patient | Healthcare professionals involved in routine asthma care (nurses, respiratory therapists, pharmacists, nurse practitioners and physician assistants). Training provided but not described | Two sessions and three telephone calls | |

COPD, chronic obstructive pulmonary disease; SDM, shared decision-making.

### Intervention fidelity

Only two studies reported intervention fidelity explicitly.[41 42] Both studies rated a specified selection of intervention audio recordings. One reported high adherence to the intervention protocol (SDM, 4/5; clinician decision-making, 3.9/5).[42] The other reported wide variation in adherence between the different intervention components, with lower delivery of cognitive-behavioural components which were described as the backbone of the intervention.[41]

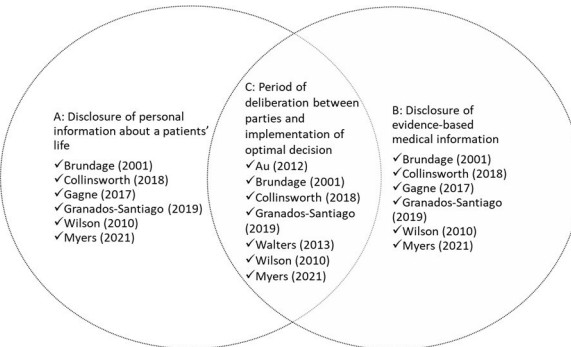

**Figure 3** Shared decision-making intervention principles aligned to The Making Informed Decisions Individually and Together model.[8]

### Attendance and attrition

For inventions involving single sessions, there was no attrition during intervention delivery.[36 37 39 43] For interventions involving multiple sessions, completion of inpatient interventions ranged from 37%[38] to 100%,[40] whereas, outpatient interventions ranged from 72%[41] to 75.5%.[42]

Results from subgroup analysis are provided in online supplemental material 5a and 5b.

### DISCUSSION

This review provides the first synthesis of studies evaluating interventions to support SDM between patients with CRD and healthcare professionals. All eight studies reported their interventions improved patient decision-making and downstream health-related outcomes. There were improvements in the quality of communication with healthcare professionals,[36] the occurrence of patient-centred communication,[36] involvement in decision-making,[41 42] decisional conflict,[39 43] health-related knowledge,[40] behavioural activation,[40 42 43] quality of life[40 42] and healthcare usage.[42] There was no one consistent outcome measure used in all studies to enable comparison across studies. Additionally, three studies measured intervention feasibility meaning they were unable to observe intervention efficacy,[37 38 43] two RCTs were judged to have a high risk of bias[38 41] and two non-RCTs were judged to have a serious risk of bias.[37 43] The most consistently reported outcomes were the downstream health-related

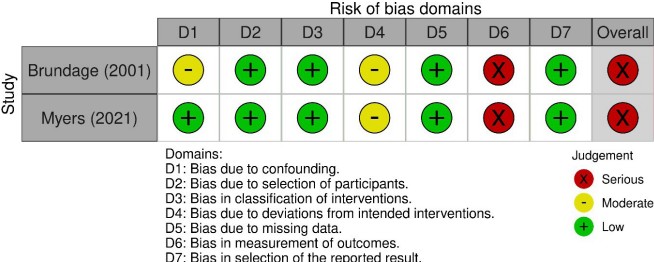

**Figure 5** Risk of bias in non-randomised controlled trial studies.

outcomes behavioural activation and quality of life. Five studies reported improvements in behavioural activation across six outcome measures.[38–40 42 43] Between group differences were only observed for two of the outcomes (nutritional status and physical activity) and this was in just one study.[40] Two studies reported improved health-related quality of life postintervention[38 42] but one of these was a feasibility study.[38]

Unlike other reviews in the CRD field,[28 29] this review found variation in the intervention components. Our appraisal showed that only five studies contained all active SDM components from the MIND-IT model (figure 3). Gagné *et al*'s[39] and Au *et al*'s[36] study did not report a period of deliberation between patients and healthcare professionals, suggesting the decision-making process was not shared, and in Walters *et al*[41] study, there was no report of patients and healthcare professionals exchanging personal and evidence-based information suggesting the decision-making process was not informed. Without adequate inclusion of all SDM components, it is difficult to conclude which intervention components support which outcomes.

Furthermore, there was little evidence in the descriptions of the interventions to show they had been developed in-line with recommended SDM intervention and user-centred design guidelines (eg, Coulter *et al* and Stacey *et al*[44 45] and O'Cathain *et al*[46]). There was also limited evidence to suggest included studies had identified their target populations' (eg, patient or healthcare professionals) needs when developing their intervention and its evaluation. For example, despite the vast array of participants' educational attainment across studies, there was little evidence studies observed this and tailored the SDM materials accordingly. This observation is of concern because lower health literacy has been shown to affect key decision-making outcomes.[47] A recent expanded model of SDM has been proposed to reflect the necessity of patients' health literacy skills as this enables them to have meaningful engagement in the decision-making process.[48] In our review, no study had considered the health literacy of their participants prior to enrolment through the use of validated measures (eg, test of functional health literacy in adults[49]), nor the use of a standardising health literacy resources such as the Plain English Campaign[50] to ensure materials were at the appropriate comprehension level and, finally, no study had considered using the low

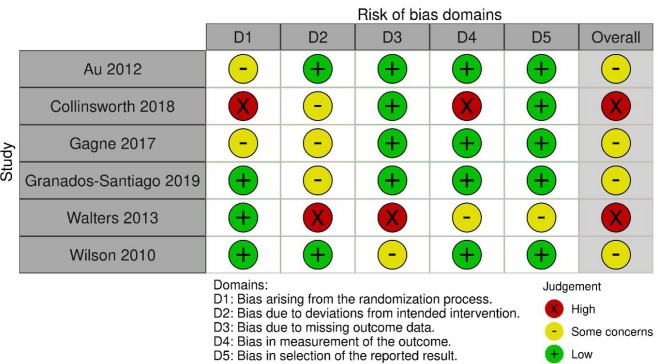

**Figure 4** Risk of bias in randomised controlled trial studies.

literacy version of the Decisional Conflict Scale,[51] despite its ease of access. Furthermore, no study reported healthcare professional-directed intervention components (eg, healthcare professional training) or used a user-centred design or implementation process.

Additionally, studies made little reference to social or familial support in the decision-making process, which is unusual as social support is often associated with improvements in self-management behaviour among patients with CRD.[52 53] Social support may contribute to informed information exchange and provide confidence in the decision-making process. Interestingly, one study specifically excluded family members or friends from the SDM consultation as they believed it would introduce bias.[37]

### Limitations of the evidence included in the review
The interpretation of these results should be considered in light of the inclusion of studies which used non-validated outcome measures. Only one study used a dose-matched intervention in the control group[41] and fidelity was only reported explicitly in two studies.[41 42]

### Limitations of the review process
The inconsistent outcome measures within this review precluded meta-analysis. Four studies had a high risk of bias[37 38 41 43] and three had low or very low quality of evidence,[37 41 43] thereby limiting the validity and generalisability of the results. The search criteria were broad, however, if studies had used alternate terminology for SDM, it is possible some may have been missed. The multi-component interventions meant intervention effects could not be discernibly attributed to the SDM component or the education and self-management components. Furthermore, two studies both provided their control groups with disease education which may have driven the positive effects observed by the control groups.[38 39] Included studies encompassed a variety of healthcare settings with decisions varying in focus from self-management to treatment decision-making and end of life care. Despite excluding studies involving hypothetical decision-making to reduce the disparity in importance attributed to the decision-making process, we cannot assume participants attributed the same importance to all decisions. Additionally, the studies length of follow-ups varied considerably, however, as the type of decisions were so varied the allocated follow-up period may have been appropriate. For example, Janssen and colleagues showed that patients with COPD change their mind about end of life care throughout the course of their healthcare journey,[54] suggesting a long-term follow-up period for end of life decision-making is appropriate. Whereas health decisions with more immediate affect (eg, acceptance of a referral to PR) may require a shorter follow-up period. As more SDM intervention studies are conducted in CRD, an updated review may well benefit from statistical subgroup analysis for studies length of follow-up and/or decision posed. Additionally, the decision to exclude qualitative and cross-sectional studies may have limited our ability to fully explore participants' experiences of the SDM process and the development and implementation of interventions.

### Implications of the results
This synthesis provides some evidence that CRD SDM interventions improve the patient decision-making process and downstream health-related outcomes. However, the evidence is weak, and the interventions at times incongruent with all active SDM components. Therefore, this limits the rigour in evaluating the impact of SDM interventions within CRD. There were also no studies exploring the efficacy of SDM in PR decision-making and implementation in practice.

Based on the available evidence, this review suggests the development of an SDM intervention to enable healthcare professionals to support COPD patients informed decision-making for PR may be beneficial. It recommends that such intervention should be evaluated within a complex intervention development and evaluation research framework. This is especially relevant given the rise in alternate PR delivery models due to the COVID-19 pandemic. Importantly, this intervention should be underpinned by SDM theoretical models (eg, MIND-IT,[8] extended SDM model involving consideration of health literacy[48]), adhere to the appropriate intervention development guidelines (eg,[44–46]) and be evaluated in a single or double-blind RCT with dose-matched controls.

**Author affiliations**
[1]Department of Respiratory Sciences, University of Leicester, Leicester, UK
[2]College of Medicine, Biological Sciences & Psychology, National Institute for Health Research (NIHR) Applied Research Collaboration (East Midlands), Leicester, UK
[3]Centre for Exercise and Rehabilitation Science, University Hospitals of Leicester NHS trust, Leicester, UK
[4]Leeds Unit of Complex Intervention Development (LUICD), University of Leeds, Leeds, UK
[5]Research Centre for Patient Involvement, Central Denmark Region and Aarhus University, Aarhus, Denmark
[6]Department of Neuroscience, Psychology and Behaviour, University of Leicester, Leicester, UK

**Acknowledgements** The authors would like to thank the UHL Library Services for their contribution to the search criteria and piloting the search. We would like to thank all study authors for sharing their data, when requested. SJS is a National Institute for Health Research (NIHR) Senior Investigator. The views expressed in this article are those of the author(s) and not necessarily those of the NIHR, or the Department of Health and Social Care.

**Contributors** ACB conceived the review. ACB, CG, LH-W, HB, NR and SJS contributed the planning, design and protocol development. ACB conducted the search in collaboration with the University Hospitals of Leicester NHS Trust Librarian. ACB and CG screened all citations. ACB contacted study authors for additional data acquisition. ACB, CG, LH-W, NR and SJS confirmed studies to be included. ACB conducted data extraction and CG conducted a 10% data quality check to ensure accuracy in this process. ACB and CG conducted risk of bias and certainty of evidence assessments. ACB conducted the narrative synthesis and ACB, CG, LH-W, NR and SJS contributed to interpretation of the results. ACB drafted the manuscript, and CG, LH-W, NR and SJS critically revised it for significant intellectual content and insight. In addition, ACB, CG, LH-W, NR and SJS gave final approval of the manuscript version for publication. ACB, CG, LH-W, NR and SJS are responsible for the overall content as guarantors.

**Funding** The National Institute for Health Research (NIHR Applied Research Collaboration (East Midlands) (NIHR ARC East Midlands) and the Centre for Exercise and Rehabilitation Science (CERS) funded this research. Award/Grant number is not applicable.

**Disclaimer** The funders had no role in the inception, data extraction or analysis phases of this review. The views expressed in this article do not communicate an official position of the University of Leicester, NIHR, CERS, University of Leeds or Aarhus University.

**Competing interests** None declared.

**Patient and public involvement** Patients and/or the public were not involved in the design, or conduct, or reporting, or dissemination plans of this research.

**Patient consent for publication** Not applicable.

**Ethics approval** Not applicable.

**Provenance and peer review** Not commissioned; externally peer reviewed.

**Data availability statement** All data relevant to the study are included in the article or uploaded as supplementary information. No additional data available.

**ORCID iDs**
Amy C Barradell http://orcid.org/0000-0002-3688-8879
Linzy Houchen-Wolloff http://orcid.org/0000-0003-4940-8835

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
