## [Reviewer comments · BMJ Open]

ARTICLE DETAILS

TITLE (PROVISIONAL)	A systematic review of shared decision-making interventions for people living with chronic respiratory diseases
AUTHORS	Barradell, Amy; Gerlis, Charlotte; Houchen-Wolloff, Linzy; Bekker, Hilary; Robertson, Noelle; Singh, Sally

VERSION 1 – REVIEW

REVIEWER	Sunjaya , Anthony The George Institute for Global Health
REVIEW RETURNED	04-Dec-2022

GENERAL COMMENTS	An interesting systematic review presenting the current gaps in resources for shared decision making in patients with CRD. Introduction - Could you clarify whether the results of recent systematic reviews on patient education materials have been evaluated to identify possibly relevant studies for inclusion? (E.g. https://www.cochranelibrary.com/cdsr/doi/10.1002/14651858.CD001431.pub5/full, https://www.ncbi.nlm.nih.gov/pmc/articles/PMC9208236/) Methods - Under inclusion criteria, please expand what the study classify and accepts as SDM and how this was done in practice to allow reproducibility.- The study design criteria in PROSPERO seems to suggest that pre-post studies can be included too. Could you clarify if any change to the criteria has been made post registration and present it as part of the Methods section?- Please confirm the references for the methods section. Ref. 30 to the Cochrane Risk of Bias tool does not seem to link to the Cochrane Handbook.- Ref. 32 also does not relate to an explanation regarding the method of narrative synthesis used for reporting the results of this study.- Was any hand searching done of the references of included studies? Results - Suggest noting the metrics used when reporting results to improve clarity. E.g. a significantly greater increase occurred in the intervention group alone [between group difference: 5.7 points; p<0.05] -> Which metric is used to ascertain this difference? How clinically meaningful is this? Discussion and Conclusion - The discussion section seems to note that 2 studies (Gagne and Au) did not report a "a period of deliberation between patients and healthcare professionals, suggesting the decision-making process was not shared". Does this meant that the two studies did not include a shared decision making intervention? Please clarify this and the definition under Methods as well as confirm whether all studies fit the inclusion criteria.
---

	- Considering the above, should studies providing patient education for COPD patients such as in https://www.ncbi.nlm.nih.gov/pmc/articles/PMC4373470/ and https://pubmed.ncbi.nlm.nih.gov/29763515/ be included in the review? Thank you.
--	--

REVIEWER	Marcussen, Michael University of Southern Denmark Faculty of Health Sciences, Department of Public Health, University of Southern Denmark
REVIEW RETURNED	19-Jan-2023

GENERAL COMMENTS	Great work. This rigorous use of the PRISMA guideline has strengthened the quality of this systematic review paper. I recommend that this review to be accepted.
--

VERSION 1 – AUTHOR RESPONSE

Introduction	Could you clarify whether the results of recent systematic reviews on patient education materials have been evaluated to identify possibly relevant studies for inclusion? (E.g. https://eur03.safelinks.protection.outlook.com/?url=https%3A%2F%2Fwww.ncbi.nlm.nih.gov/pmc/articles%2FPMC9208236%2F&data=05%7C01%7Ccab1081%40leicester.ac.uk%7Cf3c753a484aa49369cff08db08542f15%7Caebeecd6a31d44b0195ce8274afe853d9%7C0%7C0%7C638112929879021301%7CUnknown%7CTWFpbGZsb3d8eyJWljojMC4wLjAwMDAiLCJQIjoiV2luMzliLCJBTil6lk1haWwiLCJXVCi6Mn0%3D%7C3000%7C%7C%7C&sdata=yuqogAQCD%2F336V%2FgxUPycdDRNjhOnNKJCfuOGcjBRQg%3D&reserved=0)	An additional sentence added to Methods section (Information sources; see lines 198-199). “Reference lists of selected studies and relevant systematic reviews were also searched to identify further articles for inclusion.”
Methods	Under inclusion criteria, please expand what the study classify and accepts as SDM and how this was	The definition of SDM used to assess eligibility has been added to the Intervention section (see lines 172-174).

	done in practice to allow reproducibility.	“For the purpose of this review, interventions were accepted if they were patient-facing and purposely stated that were to support SDM between a patient and healthcare professional.”
Methods	The study design criteria in PROSPERO seems to suggest that pre-post studies can be included too. Could you clarify if any change to the criteria has been made post registration and present it as part of the Methods section?	An additional sentence has been added to the methods section to give examples of the types of studies included. This did include pre-post studies (see lines 190-191). “This included cluster trials, interrupted time series (ITS) studies with at least one data point before and after the intervention and controlled before-after (CBA) studies.”
Methods	Please confirm the references for the methods section. Ref. 30 to the Cochrane Risk of Bias tool does not seem to link to the Cochrane Handbook.	The Cochrane Handbook reference has been added here alongside the ROB-2 tool publication (see lines 220).
Methods	Ref. 32 also does not relate to an explanation regarding the method of narrative synthesis used for reporting the results of this study.	A further sentence has been added to this section to show the role of the reference in guiding the narrative synthesis (see lines 241-242). “Guidance on the conduct of narrative synthesis in systematic reviews [34] was consulted to guide this process.”
Methods	Was any hand searching done of the references of included studies?	Yes. An additional sentence has been added to the Methods section (Information sources) to describe this (see lines 198-199). “Reference lists of selected studies and relevant systematic reviews were also searched to identify further articles for inclusion.”
Results	Suggest noting the metrics used when reporting results to improve clarity. E.g. a significantly greater increase occurred in the intervention group alone [between group difference: 5.7 points; $p < 0.05$] -> Which metric is used to ascertain this difference? How clinically meaningful is this?	We have added metrics into the results section to make this clearer (see lines 280-375). To show the clinical significance of the results these are displayed in Table 1 alongside the properties of each measurement tool.
Discussion and conclusion	The discussion section seems to note that 2 studies (Gagne and Au) did not report a "a period of deliberation between patients and healthcare professionals, suggesting the decision-making process was not shared". Does this mean that the two studies did not include a shared decision making intervention? Please clarify this and the definition under Methods as well as confirm whether all studies fit the inclusion criteria.	Thank you for highlighting this needs to be clarified. The authors confirm that the interventions all met the eligibility criteria. A further sentence has been added to the Intervention section (see lines: 172-174). The appraisal in the discussion section is our interpretations of the interventions, using the MIND-IT model, and based upon the extracted data from the included studies manuscripts. “For the purpose of this review, interventions were accepted if they were patient-facing and purposely stated that were to support SDM between a patient and healthcare professional.”

Discussion and conclusion	Considering the above, should studies providing patient education for COPD patients such as in https://eur03.safelinks.protection.outlook.com/?url=https%3A%2F%2Fwww.ncbi.nlm.nih.gov%2Fpubmed.ncbi.nlm.nih.gov%2Fpubmed/30112929879021301%7CUnknown%7CTWFpbGZsb3d8eyJWljiMC4wLjAwMDAiLCJQIjoiV2luMzliLCJBTil6lk1haWwiLCJXVCI6Mn0%3D%7C3000%7C%7C%7C&sdata=ZSiXpv6njs6nxnABL2iX3vhiTRyTgZepcCTXE6n9efU%3D&reserved=0 and https://eur03.safelinks.protection.outlook.com/?url=https%3A%2F%2Fpubmed.ncbi.nlm.nih.gov%2Fpubmed/1529763515%2F&data=05%7C01%7Cab1081%40leicester.ac.uk%7Cf3c753a484aa49369cff08db08542f15%7Caebecd6a31d44b0195ce8274afe853d9%7C0%7C0%7C638112929879021301%7CUnknown%7CTWFpbGZsb3d8eyJWljiMC4wLjAwMDAiLCJQIjoiV2luMzliLCJBTil6lk1haWwiLCJXVCI6Mn0%3D%7C3000%7C%7C%7C&sdata=0JZxT4uu6MmX4feVKOAcqYp9lGOqJREOeYskHHVOmiM%3D&reserved=0 be included in the review?	Thank you for highlighting these studies. Following the clarifications made to the inclusion criteria section (see lines 172-174) these studies would not be accepted in this review as their purpose was not to facilitate SDM between a patient and healthcare professional.
--	---

VERSION 2 – REVIEW

REVIEWER	Sunjaya , Anthony The George Institute for Global Health
REVIEW RETURNED	03-Apr-2023
GENERAL COMMENTS	Author’s Feedback An interesting study focusing on Shared Decision Making which needs to be implemented more in practice including in the CRD space. Some comments to improve the reporting of the study further:  - Could you please expand the rationale on evaluating SDM interventions for CRD as a whole as under Introduction it was noted even for the same disease group e.g. asthma and cystic fibrosis, the evidence base found was too heterogeneous for meaningful conclusions to be reached? - While not the same, there may be benefits in looking into recent reviews on patient education materials which in many cases were part of a multicomponent intervention including face-to-face education, similar to the components defined as SDM in the study.

	E.g. https://bmcpulmed.biomedcentral.com/articles/10.1186/s12890-022-02032-9, https://doi.org/10.1002/14651858.CD001431.pub5  - Under Eligibility - could you clarify how determining an intervention supported SDM between patient and healthcare professionals was operationalised? What would happen if an intervention provided disease education, decision coaching training or SDM training but do not explicitly note SDM? - Search - there might be benefits to updating it considering the search is 1+ years to date if the author's have identified or are aware of newer studies recently published in the space. The search terms used seems to include other diseases including tuberculosis, bronchiectasis etc. Suggest noting the number of CRDs covered under Methods as it is a strength of the study and refer readers to the Supplement. - Please include under Results reference to potentially relevant studies that were excluded as noted in the PRISMA checklist. - The goal of a SDM tool is to support a patient to make a decision. Could you please expand for each included study what was the decision the tool was supposed to support? This can be part of Table 1. Were any for PR as it seems to be the topic of interest to the research group? - Under discussion, considering that most studies were feasibility/pilots and were no of high quality, would suggest clearly noting this caveat when mentioning how SDM improves quality of communication etc.
--	---

VERSION 2 – AUTHOR RESPONSE

Location within text	Reviewer query/suggestion	Action taken
Introduction	Could you please expand the rationale on evaluating SDM interventions for CRD as a whole as under Introduction it was noted even for the same disease group e.g. asthma and cystic fibrosis, the evidence base found was too heterogeneous for meaningful conclusions to be reached?	There were very few studies included in the asthma review and zero studies in the cystic fibrosis review and therefore we have added this limited dataset to the text to show another rationale for exploring the efficacy of shared decision making across chronic respiratory disease. Lines 145-150: Two systematic reviews have been published to synthesise evidence on the impact of SDM interventions on people with asthma[27] and cystic fibrosis[28] but they found the evidence base to be very limited and too heterogeneous to draw meaningful conclusions about the relationship between the interventions and improved SDM or downstream health-related outcomes and nearly all the review population were children and adolescents.
Introduction	While not the same, there may be benefits in looking into recent reviews on patient education materials which in many cases were part of a multicomponent intervention including face-to-face education,	Thank you for highlighting these reviews. https://doi.org/10.1002/14651858.CD001431.pub5 this review is already included within the introduction to highlight the body of evidence for PtDAs (Lines 116-119). Studies included in this review were also

	similar to the components defined as SDM in the study. E.g. https://bmcpulmed.biomedcentral.com/articles/10.1186/s12890-022-02032-9 , https://doi.org/10.1002/14651858.CD001431.pub5	https://doi.org/10.1186/s12890-022-02032-9 screened for eligibility for our review, however, no additional studies were added as the populations did not include participants with chronic respiratory disease (only acute). https://doi.org/10.1186/s12890-022-02032-9 We have also reviewed the included studies from this article and none were eligible because they did not meet the definition we used for shared decision making (see Lines 163-167). We have however added your review and the mention of PEMs to the introduction and inclusion criteria to show their relevance to shared decision making. Lines 110-112: There are a range of interventions currently supporting SDM in consultations, including patient education materials, for example, Patient Decision Aids (PtDAs) with or without decision coaching, and SDM training for healthcare professionals. Lines 146-148: There is a paucity of evidence to suggest patient educational materials (including PtDAs) are beneficial for reducing breathlessness and improving psychological well-being across acute respiratory diseases (Sunjaya, 2022). Lines 168-170: For the purpose of this review, interventions needed to report a patient-facing component which was designed to facilitate the decision-making process between a patient and clinician during a consultation (e.g. a patient education material).
Methods	Under Eligibility - could you clarify how determining an intervention supported SDM between patient and healthcare professionals was operationalised? What would happen if an intervention provided disease education, decision coaching training or SDM training but do not explicitly note SDM?	We have clarified and expanded this section to show that interventions needed to report a patient-facing component which was designed to facilitate the decision-making process between a patient and clinician during a consultation i.e. whilst they didn't need to use the term 'shared decision making,' they did need to state the intervention component was to support healthcare decision-making. Lines 163-167: Interventions which orchestrated SDM between patients and healthcare professionals, with or without disease education, decision coaching training, or SDM training were sought. For the purpose of this review, interventions needed to report a patient-facing

		component which was designed to facilitate the decision-making process between a patient and clinician during a consultation (e.g. a patient education material).
Methods	Search - there might be benefits to updating it considering the search is 1+ years to date if the author's have identified or are aware of newer studies recently published in the space.	We have re-conducted the search and identified no further studies to include in our analysis. We have updated the text in the following places Lines 192-193, Lines 254-255, and Figure 2.
Methods	The search terms used seems to include other diseases including tuberculosis, bronchiectasis etc. Suggest noting the number of CRDs covered under Methods as it is a strength of the study and refer readers to the Supplement.	We have added an extra sentence to highlight this. Lines 197-198: It included wide array of chronic respiratory diseases (e.g. tuberculosis, bronchiectasis, idiopathic pulmonary fibrosis).
Results	Please include under Results reference to potentially relevant studies that were excluded as noted in the PRISMA checklist.	Additional details provided. Lines 254-255: Other reasons included study design, type of intervention, and type of publication.
Results	The goal of a SDM tool is to support a patient to make a decision. Could you please expand for each included study what was the decision the tool was supposed to support? This can be part of Table 1. Were any for PR as it seems to be the topic of interest to the research group?	Thank you for this suggestion. We agree it would provide even more clarity and so have added a column into Table 1 (see Line 652). Sadly we found no interventions to support PR decision making (see Lines 453-454).
Discussion	Under discussion, considering that most studies were feasibility/pilots and were not of high quality, would suggest clearly noting this caveat when mentioning how SDM improves quality of communication etc.	Have added this caveat to the beginning of the discussion. Lines 384-386: Additionally, three studies measured intervention feasibility meaning they were unable to observe intervention efficacy[36,37,42], two RCTs were judged to have a high risk of bias [37,40] and two non-RCTs were judged to have a serious risk of bias [36,42].